# LPS-Dephosphorylating *Cobetia amphilecti* Alkaline Phosphatase of PhoA Family Divergent from the Multiple Homologues of *Cobetia* spp.

**DOI:** 10.3390/microorganisms12030631

**Published:** 2024-03-21

**Authors:** Larissa Balabanova, Svetlana Bakholdina, Nina Buinovskaya, Yulia Noskova, Oksana Kolpakova, Vanessa Vlasova, Georgii Bondarev, Aleksandra Seitkalieva, Oksana Son, Liudmila Tekutyeva

**Affiliations:** 1G.B. Elyakov Pacific Institute of Bioorganic Chemistry, Far Eastern Branch, Russian Academy of Sciences, Prospect 100-Letya Vladivostoka 152, 690022 Vladivostok, Russia; sibakh@mail.ru (S.B.); n.s.buinovskaya@gmail.com (N.B.); noskovaiulia@yandex.ru (Y.N.); sasha0788@inbox.ru (A.S.); 2Advanced Engineering School, Institute of Biotechnology, Bioengineering and Food Systems, Far Eastern Federal University, 10 Ajax Bay, Russky Island, 690922 Vladivostok, Russia; biologyvanessa@gmail.com (V.V.); bondarevgeorgii22@gmail.com (G.B.); oksana_son@bk.ru (O.S.); tekuteva.la@dvfu.ru (L.T.); 3Medical and Genetic Centre, Central Office, Laboratory of MGC “Genomed”, 8 Korolenko Str., 107014 Moscow, Russia; oxkolpakova@gmail.com

**Keywords:** *Cobetia* spp. genomes, alkaline phosphatase PhoA, PhoD, PhoX, PafA family, LPS hydrolysis, NADH pathway, oxidative phosphorylation, sulfated extracellular polysaccharides, ectoine biosynthetic gene cluster

## Abstract

A highly active alkaline phosphatase (ALP) of the protein structural family PhoA, from a mussel gut-associated strain of the marine bacterium *Cobetia amphilecti* KMM 296 (CmAP), was found to effectively dephosphorylate lipopolysaccharides (LPS). Therefore, the aim of this work was to perform a comprehensive bioinformatics analysis of the structure, and to suggest the physiological role of this enzyme in marine bacteria of the genus *Cobetia*. A scrutiny of the CmAP-like sequences in 36 available *Cobetia* genomes revealed nine homologues intrinsic to the subspecies *C. amphilecti*, whereas PhoA of a distant relative *Cobetia crustatorum* JO1^T^ carried an inactive mutation. However, phylogenetic analysis of all available *Cobetia* ALP sequences showed that each strain of the genus *Cobetia* possesses several ALP variants, mostly the genes encoding for PhoD and PhoX families. The *C. amphilecti* strains have a complete set of four ALP families’ genes, namely: PhoA, PafA, PhoX, and two PhoD structures. The *Cobetia marina* species is distinguished by the presence of only three PhoX and PhoD genes. The *Cobetia* PhoA proteins are clustered together with the human and squid LPS-detoxifying enzymes. In addition, the predicted PhoA biosynthesis gene cluster suggests its involvement in the control of cellular redox balance, homeostasis, and cell cycle. Apparently, the variety of ALPs in *Cobetia* spp. indicates significant adaptability to phosphorus-replete and depleted environments and a notable organophosphate destructor in eco-niches from which they once emerged, including *Zostera* spp. The ALP clusterization and degree of similarity of the genus-specific biosynthetic genes encoding for ectoine and polyketide cluster T1PKS, responsible for sulfated extracellular polysaccharide synthesis, coincide with a new whole genome-based taxonomic classification of the genus *Cobetia*. The *Cobetia* strains and their ALPs are suggested to be adaptable for use in agriculture, biotechnology and biomedicine.

## 1. Introduction

Alkaline phosphatases (ALPs) (EC. 3.1.3.1) belong to nonspecific metal-dependent ectoenzymes that catalyze the hydrolysis and transphosphorylation of complex phosphoric acid ethers in the environment by a mechanism involving the formation of a covalent phosphoserine intermediate and the release of inorganic phosphate (P_i_) and alcohol (or phenol) at alkaline pH values [1,2,3]. Alkaline phosphatases are widespread in nature from bacteria to humans. In mammals, ALPs are represented as a group of isoenzymes expressed in different tissues, and differ in physicochemical properties and physiological functions that are not yet fully understood [1,4,5]. The most important function of the intestinal ALP isoenzyme (IAP) is the dephosphorylation of Gram-negative bacterial endotoxins (lipopolysaccharides, LPS) and other inflammatory mediators responsible for chronic systemic diseases, as well as the maintenance of intestinal microbial homeostasis and intestinal barrier function. Thus, deficiency of IAP is responsible for several pathologies, including metabolic syndrome, liver fibrosis, coronary heart disease, osteoporosis, and ageing in general [4,6,7,8]. Increased levels of non-dephosphorylated LPS in humans correlate with an increased risk of liver and colorectal cancer [9].

The tissue-specific ALP of the columnar epithelium in animals and invertebrates has also become associated with an interaction of the host and its microbiome due to its ability to dephosphorylate a lipid A moiety of LPS [10]. Moreover, bacterial LPS and their dephosphorylation have been recognised as rather ancient functions of this family of enzymes which belong to the innate immune system of multicellular organisms, which was proven in experiments on colonization of the light organ (photophore) of the squid *Euprymna scolopes* by its symbiotic luminescent bacterium *Vibrio fischeri* [11]. An increase in the level of dephosphorylated LPS by sunset is a signal for the bacteria to increase their population size and, accordingly, the intensity of photophore luminescence. At the same time, the LPS dephosphorylation protects the invertebrate from excessive inflammation and destruction of its tissues under the influence of endotoxin, LPS with phosphorylated lipid A [11].

In bacteria, ALPs play a major role in utilizing organic phosphates as an alternative source of the vital macronutrient, phosphorus (P), in P_i_-deficient environments [12,13,14]. The bacterial ALPs catalyze the hydrolysis of sugar phosphates, DNA and RNA (5′-, 3′-ends), nucleotide mono-, di-, and triphosphates (dNMP, dNDP, dNTP), lipid phosphatidates, polyphosphates (polyP), and pyrophosphates, which are prevalent in environments rich in organics [3,15,16,17]. However, some ALPs were found to catalyze the cleavage of sulfate and phosphate di- and triesthers, including neurotoxins with P-O-C bonds, due to their evolutionarily catalytic proficiency, and are thus ecologically relevant [13,18,19]. In addition, ALPs non-specifically dephosphorylate some proteins, many of which are part of cell signalling transduction [20,21]. Finally, microbial ALPs are of global biogeochemical importance, being among the most abundant enzymes in soils and the world’s ocean [22,23,24,25,26]. Alkaline phosphatases are used by bacteria competing with fungi to regulate their metabolic pathways [27]; to induce bacterial biofilms and growth and mineralization of invertebrate exoskeletons [15,28,29]; and to participate in the remediation of heavy metals and organic pollutants [13,14,30]. Alkaline phosphatases are widely distributed among marine bacteria that extract P_i_ from the dissolved organic P-containing compounds in the global ocean [12,13,14,22,25,31]. In addition, marine bacteria and diatom algae can accumulate P_i_ and polyP, with their subsequent release from the cells into the environment and a concomitant increase in the level of phosphatase activity [12,32]. Thus, microorganisms take an active part in the enzymatic induction of nucleation and crystal growth of such minerals as apatite and phosphorites [24,25,26].

Currently, four large structural families of prokaryotic ALPs are known, namely: PhoA, PhoD, PhoX, and P_i_-irrepressible PafA [14,22,23,26]. They differ from each other in structure, mechanism of enzymatic action, subcellular localization, substrate specificity and dependence of the manifestation of their activity on various metal ions, temperature, and pH ranges [14,31,33]. Multicellular organisms produce the Mg^2+^/Zn^2+^-dependent ALPs structurally belonging to the PhoA family, which includes ALPs of the model microorganism *E. coli* and mammals, which are better characterized and considered as classical ALPs [1,2,16,29]. Despite the conservation of main catalytic mechanisms and characteristics, mammalian ALPs have higher specific activity and effectiveness, and lower values of the Michaelis constant (*K*_M_), as well as a more alkaline optimum pH compared to bacterial ALPs [1]. However, the enzymes of some marine bacteria, for example, of *C. amphilecti* KMM 296 (CmAP), have catalytic characteristics comparable to mammalian ALPs and a significant enzymatic activity in the monomeric state that makes them suitable for use in biotechnology [33].

In this study, we used a quantitative phosphorus assay to determine the enzymatic activity of *C. amphilecti* KMM 296 alkaline phosphatase CmAP on *E. coli* LPS. We identified the ALP structural family of the enzyme CmAP, as well as all homologous CmAP-like structures and non-homologous ALPs in all genomes of the *Cobetia* spp. strains available in the NCBI database. To perform phylogenetic analysis for the *Cobetia* spp. ALPs, the known LPS-dephosphorylating ALPs of humans and invertebrates with the established function of modulating their microbiome were used. The species-specific identification of the *Cobetia* isolates was done based on the whole-genome ALP contents, and the biosynthetic gene profiles and identities, to determine the differences between the PhoA-containing and non-containing bacteria. Analysis of the gene *phoA* localization in the *Cobetia* spp. genomes was carried out to suggest its function and participation in metabolic pathways.

## 2. Materials and Methods

### 2.1. Recombinant Production of the C. amphilecti KMM 296 Alkaline Phosphatase CmAP

The recombinant protein CmAP was produced in the *E. coli* strain Rossetta DE3 using the recombinant plasmid Pho40 based on the pET-40b(+) vector (Sigma-Aldrich, Burlington, MA, USA) containing the full-length coding sequence of the mature alkaline phosphatase CmAP from the marine bacterium *C. amphilecti* KMM 296, as described earlier [33]. For heterologous expression, the competent cells of *E. coli* Rosetta (DE3) were transformed by the recombinant plasmids Pho40 and pET-40b(+) as a control (Appendix A), and grown at 37 °C on agar LB medium containing 25 µg/mL of kanamycin overnight. The recombinant clones were grown in 25 mL of the liquid LB medium containing 25 µg/mL of kanamycin at 200 rpm for 16 h at 28 °C. The cell cultures were placed in a fresh LB medium (1 L) containing kanamycin (25 µg/mL) and incubated at 37 °C on a shaker at 200 rpm until the optical density at 600 nm was 0.6–0.8. After that, 0.2 mM isopropyl-β-D-thiogalactopyranoside (IPTG, Sigma-Aldrich, Burlington, Massachusetts, US) was added to induce the recombinant gene expression, and further incubated at 16 °C (200 rpm) for 16–18 h. Cells were pelleted via centrifugation at 4293× *g* for 15 min at 8 °C, suspended in 250 mL of buffer A containing 20 mM Tris-HCl (pH 9.0), 5 mM imidazole, 0.5 M NaCl, 20% glycerol (Sigma-Aldrich, USA), and subjected to an ultrasonic treatment by Bundeline SONOPULS HD 2070 (Berlin, Germany) to provide a complete release of the soluble recombinant protein from the *E. coli* periplasmic space.

After centrifugation of the disintegrated lysate at 32,467× *g* for 20 min, the supernatant was saturated with dry ammonium sulfate up to 30%. The precipitate was removed by centrifugation (15 min at 32,467× *g*, 4 °C). The supernatant was collected and saturated with ammonium sulfate up to 70% over the course of an hour, resulting in precipitation of the recombinant protein. The precipitate was centrifuged for 15 min at 32,467× *g* at room temperature and re-suspended in buffer A. The resulting supernatant was applied to a metal affinity resin (Ni^2+^-IMAC-Sepharose, V column = 160 mL, GE Healthcare, Chicago, IL, USA), equilibrated in the same buffer at the rate of 1 mL/min. The column was washed with buffer A, after which the recombinant protein was eluted with a linear gradient of 0.005–0.5 M imidazole in buffer A at an elution rate of 2 mL/min. After that, dialysis was performed for 12 h against buffer A containing 50% glycerol (Sigma-Aldrich, USA). The His-tag of the recombinant protein was removed during 12 h of incubation with 1–2 units of enteropeptidase (L-HEP) per 1 mg of the recombinant protein with stirring. After chromatography on the metal affinity sorbent, the unbound fractions were collected and applied to Source 15 Q ion exchange sorbent (column V = 8 mL, GE Healthcare) at a rate of 5 mL/min. Elution was carried out with a gradient of 0.15–1.0 M NaCl in buffer A (without the salt) at the rate of 2 mL/min (Appendix A). Fractions of the target protein were identified using *p*-nitrophenyl phosphate (*p*-NPP) by the presence of ALP activity and by the molecular weight determined by SDS-PAGE electrophoresis [34].

The recombinant CmAP (Appendix A) was applied to a 0.5 mL High-Capacity Endotoxin Removal Spin Column (Thermo Scientific^TM^ Pierce^Tm^, Waltham, MA, USA) and the recombinant *E. coli* LPS absence in the final preparation of CmAP was checked using a Chromogenic Endotoxin Quantification Kit (Thermo Scientific^TM^).

### 2.2. Alkaline Phosphatase Activity Assay

The standard assay for ALP activity was carried out in 500 μL of the reaction mixture containing 2 mM *p*NPP (Sigma-Aldrich, USA) in 0.1 M Tris–HCl buffer, pH 10.0, 0.2 M KCl at 37 °C for 30 min. The release of *p*-nitrophenol (ε = 18.5 mM/cm) was monitored at 405 nm after the addition of 2 mL 0.5 M NaOH. One unit of the ALP activity was defined as the quantity of the enzyme required to release 1.0 μM of *p*-nitrophenol from *p*-NPP in 1 min. The specific activity was calculated as units (U) per 1 mg of protein [33].

### 2.3. Dephosphorylation Activity Assay towards E. coli LPS

The smooth-form LPS (S-LPS) from *E. coli* serotype 055:B5 (Sigma-Aldrich, USA) was used as a substrate containing phosphoester bonds for the enzymatic reaction. The LPS samples (0.1 and 0.2 mg/mL) were dissolved in 0.1 M Tris-HCl buffer, pH 7.7, containing 0.1 M KCl in three ways: (1) incubation at 24 °C for 12 h (sample N 1); (2) incubation at 37 °C for 12 h (sample N 2); (3) incubation at 24 °C for 12 h in the same buffer with the addition of triethylamine (TEA, 1µL/mL), an organic dispersant solvent, to pH 10.0 (sample N 3). The LPS sample N 4 was prepared in 0.1 M Tris-HCl buffer, 0.1 M KCl, pH 10.0, at 24 °C for 12 h.

The enzymatic activity of the recombinant alkaline phosphatase CmAP (0.3 mg/mL, 2300 *p*-NPP U/mg), with the use of LPS as substrate, was determined by the quantitative analysis of phosphorus remaining in the LPS samples after hydrolysis. An aliquot (10 µL) of the enzyme dissolved in 1 M DEA buffer, pH 10.3, to final concentrations 0.006, 0.012, and 0.024 mg/mL and various LPS samples (240 µL) were mixed in glass vials with caps (total volume 250 µL) and incubated at 37 °C with stirring for 1 h. After the end of the reaction, the samples were transferred into dialysis tubes (with a pore size of 3000 Da) and dialyzed against distilled water for 2 days in a cold room for the removal of free phosphorus.

For quantitative determination of phosphorus in the LPS solution before and after treatment with the alkaline phosphate CmAP, a universal molybdate reagent was used [35], for which a working solution was prepared: 26 mL of 1 N sulfuric acid was added to 5.5 mL of the initial reagent and the volume of distilled acid was adjusted with water up to 100 mL. An aliquot of the LPS solution (70 µL) was taken into a test tube and evaporated to dryness in a heating oven at 100 °C. Then, 0.05 mL of 72% perchloric acid was added to the dry residue and burned in a duralumin block at 180–200 °C for 20 min. After cooling, 0.45 mL of the working reagent was added to the test tubes. The mixture in the test tube was thoroughly mixed using a vortex, and the test tubes were placed in a boiling water bath for 15 min. After the formation of phosphomolybdenum blue, the tubes were cooled and the optical density of the samples was measured in a quartz cuvette (l = 1 cm) at 815 nm on a Specol spectrophotometer (Carl Zeiss, Jena, Germany). For each sample, three parallel measurements were made. For each measurement, a control sample was used (buffer without LPS with the same concentration of the enzyme CmAP), the absorbance of which did not exceed 0.04–0.05 optical density units.

A calibration curve for determining phosphorus in the LPS samples was drawn using monosodium phosphate (NaH_2_PO_4_ × 2H_2_O) (AppliChem, Darmstadt, Germany) dissolved in water (stock solution—7.8 mg in 250 mL) in the concentration range 0.03–0.3 μg/mL.

The data obtained were visualized using the Python library Seaborn 0.13.2.

### 2.4. Phylogenetic and Biosynthetic Gene Cluster Analyses

For identifying the homologues of the *C. amphilecti* KMM 296 alkaline phosphatase CmAP (accession no. KGA01942), global blast, blastn, and tblastn (Available online: https://blast.ncbi.nlm.nih.gov/Blast.cgi, accessed on 18 August 2023) were performed.

For the ALP phylogeny, 36 publicly available *Cobetia* spp. genomes were used to search for the genes (CDS) encoding for the homologues of the *C. amphilecti* KMM 296 alkaline phosphatase CmAP (accession no. KGA01942), as well as other genes and gene products related to the functional annotation “alkaline phosphatase” (Appendix A, sheet “ALP types”). Using our own R scripts for processing tabular data (libraries: rtackleyer, readr, plyr, dostats, rtracklayer, sybil, data.table), the GTF files for all *Cobetia* strains were analyzed to create a list of genes described as “alkaline phosphatase” (Appendix A, sheet “ALP types”). In the GTF files, there are 3 kinds of alkaline phosphatase descriptions: “alkaline phosphatase”, “alkaline phosphatase D family protein”, and “alkaline phosphatase family protein”. The protein IDs found were used for multiple alignment and phylogenetic tree construction. In addition, the reference ALP proteins of the family PhoA were included in the analysis, namely: *Homo sapience* (NP_001622.2, P09923.2), *E. scolopes* (AER46070, AER46069), *Moritella* sp. 5 (QUM82918), *Vibrio* sp. G15-21 (Q93P54), and *E. coli* (NP_414917, NP_311634, NP_417233). The reference ALP proteins of the family PafA were of *Flavobacterium* spp.: WP 073408805, WP 011921542, WP 149206384, WP 198858267, WP 091133708. Gene products with an annotation “alkaline phosphatase D family protein” were taken as the family PhoD ALPs (Appendix A). Unclassified proteins with a functional annotation “alkaline phosphatase family protein” were referred to the family PhoX or PafA, which were manually checked and compared with literature data (Appendix A). Using gene_neighbor.R and the Integrative Genomic Viewer (IGV) browser (available online: IGV; accessed on 23 July 2023), the *metG* gene neighbouring clusters containing the CmAP homologues (KGA01942) were identified in all GTF to analyse their location in the chromosome of each strain (Appendix A, sheet “ALP neighbours”).

The evolutionary history was inferred by using the Maximum Likelihood method and Jones et al., w/freq. model [36]. The tree with the highest log likelihood (−22,741.47) was selected. The initial trees for the heuristic search were obtained automatically by applying Neighbor-Join and BioNJ algorithms to a matrix of pairwise distances estimated using the JTT model, and then selecting the topology with a superior log likelihood value. The analysis involved 137 amino acid sequences. There was a total of 880 positions in the final dataset. Evolutionary analyses were conducted in MEGA 11 [37].

To search for metabolic pathways involving the homologues of the *C. amphilecti* KMM 296 alkaline phosphatase CmAP (accession no. KGA01942), the *gapseq* software (https://github.com/jotech/gapseq, accessed on 23 July 2023) was used to build genome-wide metabolic network models using a modified automatically updated ModelSEED biochemistry database [38]. Prediction of pathways, transporters, and complexes is based on a protein sequence database derived from UniProt and TCDB.

To visualize the biosynthetic gene cluster (BGC) synteny, clinker, a Python-based tool, and clustermap.js, a companion JavaScript visualization library, were applied. Together, they allowed automatic generation of accurate interactive gene cluster comparison figures directly from sequence files [39]. Biosynthetic gene clusters were identified using antiSMASH (https://antismash.secondarymetabolites.org/, accessed on 9 December 2023), after which the resulting .gbk sequences were aligned using clinker. The cluster alignments created using clinker were visualized using clustermap.js and edited using Inkscape [40].

## 3. Results and Discussion

### 3.1. Dephosphorylation Activity of the Recombinant Alkaline Phosphatase CmAP towards E. coli LPS

The *E. coli* LPS was used as the P-containing substrate for the *C. amphilecti* KMM 296 alkaline phosphatase CmAP (accession no. KGA01942), with high ALP-specific activity and catalytic efficiency relative to other bacterial homologues, to study its physiological function and substrate specificity in nature [3,10,15,28,33].

Bacterial LPS recognition and dephosphorylation is stated to be a very ancient function of ALPs and belongs to the innate immune system of multicellular organisms [10]. Alterations in the expression and activity of LPS-dephosphorylating ALP of the light organ (photophore) of the squid *E. scolopes* regulate colonization and abundance of its luminescent symbiont *V. fischeri* [11]. The increase in the level of dephosphorylated LPS by sunset is a signal for the bacteria to increase population size and consequently photophore luminescence intensity, and simultaneously protect the invertebrate from excessive inflammation and destruction of its tissues by the endotoxic phosphorylated lipid A [11]. The possible effect of the bacterial ALP on LPS has not been investigated except for the finding that exogenous *E. coli* ALP, which was applied to mice infected with *Pseudomonas aeruginosa*, positively modulates the growth of commensal bacteria in the animals [41]. This probably led to increased competitiveness of their microflora with the infectious agent and reduced its production of enterotoxins [41].

LPS is the main component of the outer membrane of Gram-negative bacteria, occupying about 70% of the surface [42]. The complete structure of the amphiphilic LPS molecule (S-form of LPS), a characteristic of most wild-type bacterial strains found in nature, consists of three parts: hydrophobic lipid A, core oligosaccharide, and O-polysaccharide chain. The O-polysaccharide is attached to the core oligosaccharide, which in turn is bound to the lipid A moiety. Lipid A in most of the bacteria studied is a dephosphorylated *β*-1,6-linked glucosamine disaccharide (diaminogenciobiose) bearing higher fatty acid residues. In addition to the S-form of LPS, bacteria also synthesize core-defective LPS molecules (R-chemotypes), which have lipid A and a core part of different lengths. The core-defective structures are labelled as Rb-Re-chemotypes, and the structures with a complete outer core as Ra-chemotypes [43,44]. In the LPS micelles formed in aqueous solutions, lipid A is located inside the micelles, and the O-polysaccharide, constructed in most cases from hydrophilic monosaccharide residues, as well as in bacterial cells, is directed towards the aqueous phase. Therefore, the solubility of LPS depends on the length of the O-polysaccharide [42,44]. However, the morphology and particle sizes of LPS preparations are not only dependent on the chemical structure of the LPS samples but also on concentration, ambient conditions, and the techniques applied [43].

Thus, the amount of detectable phosphorus in the LPS sample solubilized in 0.1 M Tris-HCl buffer, pH 7.7, increased 6-fold after reducing the LPS concentration from 1 to 0.2 mg/mL and increasing the incubation temperature to 37 °C (sample N 2) (buffer A of Figure 1). It was comparable with the results on the LPS samples N 4 (59 and 62% removed P_i_) prepared at 24 °C in the same buffer but at a strongly alkaline pH (buffer B of Figure 1), in which the CmAP activity towards *p*-NPP is optimal [33].

The quantity of removed P_i_ was significantly increased after pre-incubation of the LPS samples (sample N 3) with a surfactant TEA at 24 °C and, consequently, alkalization of the incubation medium to pH 10.0 compared to the LPS sample N 1 (without the addition of TEA). The maximum phosphorus content was determined in LPS sample N 3 (buffer A of Figure 1). Thus, the CmAP activity towards LPS samples was much more dependent on the LPS desegregation level than the pH value in the conditions used for the enzymatic reaction [33,43].

As expected, the maximum dephosphorylation activity of the recombinant enzyme CmAP was observed against the highly disaggregated LPS sample N 3, for which the amount of hydrolyzed phosphorus was 81.2% of the control sample (not treated with alkaline phosphatase CmAP) (buffer A of Figure 1). The interaction between the substrate-binding site of the enzyme and LPS is impeded in the case of larger particle sizes due to their poor solubility in the absence of surfactant, as well as at a lower pH value (LPS sample N 2) and temperature (LPS sample N 1 and 4) (Figure 1). Similarly, the mammalian LPS-binding protein (LBP) bioactivity is governed by the interaction with particular functional groups within the lipid A backbone, and is dependent on the lipid A fluidity and aggregation state [43].

Thus, the alkaline phosphatase CmAP of the marine bacterium *C. amphilecti* KMM 296 exhibits enzymatic activity towards the *E. coli* LPS, which depends largely on the degree of availability of phosphate residues in the LPS molecule in a solution (disaggregation state). The enzyme CmAP at the concentration of 0.012 mg/mL can dephosphorylate LPS at a concentration of 0.2 mg/mL almost completely if it is pre-incubated for 12 h at 24 °C in the presence of dispersant TEA (Figure 1).

The specificity of the marine bacterial ALP CmAP towards LPS is similar to the property of epithelial enzymes of eukaryotes, in particular the human isoenzyme IAP, which dephosphorylates LPS of human intestinal and pathogenic microflora to prevent inflammation and persistence of bacteria and endotoxins in the bloodstream [4,6,7,8,10]. The findings on the LPS-dephosphorylation activity of the marine bacterial enzyme CmAP may provide a promising basis for the development of a novel therapeutic approach to neutralize the effects of bacterial endotoxins.

### 3.2. Multiplicity of Alkaline Phosphatase Phylotypes in the Species Cobetia amphilecti

#### 3.2.1. Structural Classification of the Bacterial Alkaline Phosphatases

Bacterial ALPs can be divided by the amino acid sequence (primary structure) into three main structural families: PhoA, PhoD, and PhoX, belonging to the COG1785, COG3540, and COG3211 proteins, respectively, which originate from different ancestral genes according to the classification of the Clusters of Orthologous Genes (COG) database [26]. The structure of the PhoA family enzymes was the first to be studied, as the classical phosphomonoesterase of *E. coli* belongs to it [10]. Subsequently, other enzymes non-homologous to the *E. coli* alkaline phosphatase, but with similar physiological functions, were discovered in environmental bacteria [13,14,22,23,31]. The high variability of protein sequences in the alkaliphilic enzymes is characteristic both for representatives of different families and within each family, depending on the taxonomic affiliation of their microbial producers [23,26]. Despite the low sequence identity among representatives of the PhoA, PhoD, and PhoX families, their main common property is the production of P_i_ during the depletion of its stores in the environment, and, consequently, their expression and enzymatic activity is inhibited by high P_i_ concentrations according to a feedback response [12,14]. However, unlike the Mg^2+^-activated and Zn^2+^-containing phosphomonoesterases of the PhoA family, the members of PhoD and PhoX families exhibit their maximal activity in the presence of Ca^2+^/Co^2+^ ions against a wider range of substrates, both phosphomono- and diesters [19,23,31]. In addition, some alkaline phosphatases/phosphodiesterases PhoD contain Fe^3+^ in their bimetallic active centres instead of Zn^2+^ [31,45]. Such an unusual architecture of the active centre and its Fe^3+^-specificity could have evolved in the zinc-deficient as an adaptation of the bacterium to survive in a particular environment. This property may have been acquired by soil bacteria from a plant purple acidic phosphatase PAP [45].

Some authors include in the classification of bacterial ALPs a fourth family of constitutive highly active Ca^2+^-dependent enzyme PafA, with a broad substrate specificity and unknown metabolic function, common in the genomes of flavobacteria (mainly *Bacteroidetes*) associated with plant rhizospheres, because their expression and enzymatic activity are not inhibited by the reaction product, P_i_, and are not controlled by known regulators unlike other ALPs [14,26]. The presence of non-inducible and unrepressed PafA alkaline phosphatase in flavobacteria has been shown to promote rapid remineralization of various organophosphates and P_i_ accumulation, which enables a secondary growth of other bacterial species in microbiomes [14]. In addition, the PafA-like ALPs can be active against phosphodiesters, expanding the role of these enzymes in nature [14,18]. However, some representatives of the PhoD, PhoX, and PhoA families may exhibit similar properties [13,14,18,19]. The highly active and catalytically efficient enzyme CmAP isolated from the marine bacterium *C. amphilecti* KMM 296 was also not inhibited by a large concentration of P_i_ [15,33]. The recombinant PhoA analogue from a marine bacterium *Alteromonas mediterranea* showed a broad substrate specificity towards phosphodi-, phosphotriesters, and sulfates at low substrate concentrations, whereas the enzyme exhibited a high phosphomonoesterase catalytic efficiency at high substrate concentrations [13]. Thus, the structural classification of the ALP enzymes only reflects their belonging to different homologous lineages, which descended from different ancestral genes that evolved independently to perform the same functions in the organism and/or microbiome [22,23,26].

#### 3.2.2. Searching for Homologues of the *C. amphilecti* KMM 296 Alkaline Phosphatase CmAP

According to the blast-based searching against the sequence query KGA01942 encoding for the *C. amphilecti* KMM 296 alkaline phosphatase CmAP, it is a homologue to the PhoA *E. coli* ALP (Appendix A). The multiple alignments showed that the PhoA sequences of *E. coli* differ significantly from the PhoA of *C. amphilecti*, with an overall identity of 32.21% and a cover of 61%. The search for the KGA01942 protein among the *Cobetia* spp. genomes (Appendix A) identified eight homologous sequences in 35 strains, indicating that it is characteristic of the strains of the subspecies *C. amphilecti* (Appendix A) according to the gene-specific and genome-based classification of the *Cobetia* genus as described earlier [46,47]. Although both groups of the strains, *C. amphilecti* and *C. litoralis*, belong to the same phylotype in terms of the whole genome characteristics [47], the species is divided into two subspecies based on the content of alkaline phosphatases (Table 1 and Appendix A). This divergence probably emerged when the parent population changed its lifestyle from free-living to host-associated [46].

Remarkably, the mussel-associated strain *C. amphilecti* KMM 296 and the coral-associated type strain *C*. *amphilecti* NRIC 0815^T^, as well as other available *C. amphilecti*-like isolates, have been found to possess five proteins with the functional annotation “alkaline phosphatase”, including the PhoA family ALP, in contrast to four ALPs found for the *C. litoralis*-like strains (Table 1, Appendix A). In addition, the *phoA* gene was found in the type strain *C. crustatorum* JO1^T^ from a shrimp and two strains *C. crustatorum* SM1923 and *Cobetia* sp. QF-1 (Appendix A), which are also related to the *C. crustatorum* species [47]. However, the *C. crustatorum* species are phylogenetically distant from other *Cobetia* species, therefore they are best classified using 16S RNA gene analysis, unlike the closely related and undistinguishable isolates of *C. amphilecti* and *C. litoralis* or *C. marina* and *C. pacifica*, respectively [46,47].

Nucleotide substitutions in the PhoA family coding sequences were detected in the *C. amphilecti*-like genomes (~1%) and the genomes of *C. crustatorum* JO1^T^ and SM1923 (~11%) (Table 2). The substitutions do not hit the active sites of the enzymes except for *C. crustatorum* JO1^T^, which has a down start (+45 amino acids) and the missing key position 21D (a catalytic Asp), indicating a non-functional homologue of the enzyme CmAP [33]. However, the exact metabolic function of the enzyme CmAP (KGA01942) and its homologues were not identified by *gapsec* except for the pathway producing NADH from the consumed NADPH, mostly associated with the intracellular 5′-nucleotidase (*surE*), probably due to the low identity of the model enzymes, the participation of which in metabolic reactions is verified [38]. *Gapseq* detected pathways associated with ALP in all strains because the blast-based homologues of the desired enzyme are found in the genomes of all bacteria studied, but they are short (<100 amino acids) and with a low homology (Expect > 1, Identities < 50%).

Using the IGV browser, an unnamed gene with an alkaline phosphatase product homologous to the *C. amphilecti* KMM 296 CmAP (KGA01942) was identified in the GTF files of ten *Cobetia* strains, predominantly located next to the methionine-tRNA ligase gene *metG* (+/−2 positions) (Figure 2). In this regard, we decided to isolate the region centred on the *metG* gene. Appendix A (sheet “ALP neighbours”) was created based on the results of the analysis. Each cell in Appendix A contains a gene and a product in the format “gene|product” (e.g., “*metG*|methionine-tRNA ligase”). The CmAP-like gene (KGA01942) is presented in the table as “NA|alkaline phosphatase” because the gene name is not labelled on the map (without functional annotation). However, the non-homologous ALP of *C. pacifica* NRIC 0813^T^ is also located in a similar *metG*-cluster (Appendix A). Although only 10 strains have the CmAP-like gene in the immediate neighbourhood of the *metG* gene (Figure 2, Appendix A, sheet “ALP neighbours”), similar extended *metG*-regions for the different *Cobetia* spp. strains are observed. It should be noted that in the genomes of *Cobetia* sp. UCD-24C (GCF_001306765.1) and *C. crustatorum* JO1^T^ (GCF_000591415.1), the CmAP-like genes (e.g., KGA01942) are not located in the neighbourhood of the *metG* gene (Appendix A). In the genomes of *Cobetia* sp. UCD-24C (GCF_001306765.1), *C. crustatorum* JO1^T^ (GCF_000591415.1), *C. marina* NBRC 15607 (GCF_006540105.1), and *Cobetia* sp. 10Alg146 (GCF_029846385.1), the *metG* neighbour genes differ significantly (Appendix A).

In addition to *metG* and unidentified alkaline phosphatase (NA), in the immediate environment there are electron transport complex subunits RsxD, RsxC, RsxB, RsxA, DUF3465 domain-containing protein (Si-specific NAD(P)(^+^) transhydrogenase), TerB family tellurite resistance protein, cell wall hydrolase (NA), RNA pseudouridine synthase (NA), double *apbC* genes encoding for iron-sulfur cluster carrier proteins ApbC playing a role in regulating NADH oxidation, dCTP deaminase, and autotransporter domain-containing SGNH/GDSL hydrolase family protein (protease and lipase related) (Figure 2, Appendix A, sheet “ALP neighbours”). Such gene clusters are essential in electron transfer reactions, respiration, DNA repair, and gene regulation and, consequently, may be related to the rearrangement of the cell wall and cellular phenotypes under oxidative stress or iron/sulfur limitation in both Gram-negative and Gram-positive bacteria [48,49]. These facts might explain the biofilm-degrading behaviour of many different bacteria after their treatment with the PhoA alkaline phosphatase CmAP solution in a dose-dependent manner [28]. Besides P-nutrient scavenging under P_i_ deficiency, PhoA was found to constrain secondary metabolite biosynthesis and cell division in the diatom algae *Phaeodactylum tricornutum*, under P-replete conditions. These functions have important implications in maintaining metabolic homeostasis and preventing premature cell division [50].

Thus, the PhoA alkaline phosphatase in some *Cobetia* species may be suggested to participate in cellular energy metabolic pathways, particularly, oxidative phosphorylation (Figure 3).

As predicted by *gapsec*, the CmAP-like PhoA, together with the energy-dependent proton-translocating NAD(P)(^+^)-transhydrogenase, manipulate the ratios of cellular NADH/NAD^+^ and NADPH/NADP^+^ to maintain cellular redox balance through the following reaction: H ^+^_out_ + NADP^+^ + NADH ↔ H ^+^_in_ + NADPH + NAD^+^ [51,52]. In most metabolic conditions, NADP(H) is used in reductive reactions and the NADP^+^:NADPH ratio is typically < 1, whereas NAD(H) is associated mainly with oxidative catabolic reactions with the NAD^+^:NADH ratio > 1 [53]. These processes are coupled with pumping of protons (H^+^) from the extracellular space into the periplasm, generating an electrochemical gradient which drives ATP and GTP production by synthases (Figure 3). The electron transport chain (ETC) inevitably produces by-products, reactive oxygen species (ROS). A steady level of ROS is beneficial and required for many biological processes. As signaling molecules, ROS play an important role in cell proliferation, hypoxia adaptation, and cell fate determination [54]. Under stressed states, such as hypoxia, NAD(P)^+^-transhydrogenase increases NADH/NAD^+^ leading to dysfunction of the ETC complex I and additional ROS production (oxidative stress). NADPH serves as an anti-oxidant; therefore, anti-oxidative defence strategy aims at converting the pro-oxidant NADH into NADPH [51,54,55,56].

However, under an excess of phosphate nutrition (P-replete conditions), diatom alga increased the alkaline phosphatase PhoA expression with a concomitant decrease in the biosynthesis of secondary metabolites with antioxidant functions, such as tetrapyrroles and fatty acids, energy generation, and iron uptake [50]. This may indicate that alkaline phosphatase directly or indirectly shifts the transhydrogenase reaction towards NADH formation, which is predominant at the stationary phase of cell growth where NADPH is not required [57].

In general, the highly active P_i_-irrepressible periplasmic CmAP-like enzyme, which may quickly deliver P_i_ for biosynthesis of high-energy purine nucleotides (*metG*, *purM*, *purN*) and related cofactors, contributes to regulation of DNA replication and the cell cycle (DnaA related gene *hda*) in response to environmental changes, which might involve competition (*dcd*, *terB*), high salinity, autotrophic or mixotrophic growth on inorganics, etc. (Figure 2 and Figure 3) [57,58,59,60]. These environmental changes are probably related to aeration conditions and CO_2_ level, since the electron transport chain (*rsxABCDG* or *rnfCDGEABF*) and Fe-S carrier (*apbC*) protein genes (Figure 2 and Figure 3) for energy conservation are found in facultative anaerobic bacteria (*E. coli*, *Rhodobacter capsulatus*, *Vibrio cholera, Salmonella enterica, Erwinia* spp.), anaerobic acetogenic or sulfate-reducing bacteria (*Clostridium* spp., *Desulfovibrio* spp.), and methanogenic archaea [58,61,62].

In addition, marine bacteria have been recently found to continuously acquit and accumulate P_i_ into the periplasm across the outer membrane (OM) in the proton motive force (PMF)-dependent manner, and this can be enhanced by light energy, where the periplasmic P_i_ anions pair with chemiosmotic cations of the PMF to influence the periplasmic osmolarity of marine bacteria [32]. Therefore, the highly active P_i_-irrepressible CmAP-like alkaline phosphatase PhoA is a rather multifunctional enzyme for marine bacteria.

#### 3.2.3. Phylogenetic Analysis of Alkaline Phosphatases in the Genus *Cobetia*

The phylogenetic analysis of identified *Cobetia* spp. ALPs revealed that each *Cobetia* strain contains from two to five ALPs belonging to different structural families, of which two different PhoD or two different PhoX structures are necessarily present (Table 1 and Appendix A, Figure 4).

Each PhoD or PhoX structure from one strain seems to belong to a different lineage, but to be species-specific according to their clusterization into the species-based groups within one homologous lineage (Figure 4). For example, the strain *C. amphilecti* KMM 296 has two PhoD sequences under the accession numbers WP_043333989.1 and WP_052384691.1 (Table 1), which are respectively located on the left and right PhoD branches of the phylogenetic tree, indicating their origination from different parent genes (Figure 4). The *C. marina* strains also possess two PhoD genes located in different PhoD branches, each of which differs from the sequences of *C. amphilecti* of the same homologous lineage (Figure 4). Moreover, the addition of the referent sequences from *Flavobacterium* spp. belonging to the novel P_i_-irrepressible ALP family PafA [14,26] allowed the identification of Paf-like proteins in the *Cobetia* isolates, which were clusterized in separate branches (Figure 3) consisting of other ALP enzymes that coincide with the unclassified alkaline phosphatases from Appendix A (Appendix A, sheet “ALP types”, coloured in red). Remarkably, the PafA family was found only in the genomes of the species *C. amphilecti*, including both subspecies, *C. amphilecti* and *C. litoralis* (Table 1 and Appendix A, Figure 4). This indicates that the *C. amphilecti* strains may colonize plant roots similarly to PafA-containing flavobacteria [14]. Indeed, the strain *Cobetia* sp. UCD-24C was isolated from *Zostera* sp. Roots, and was found to possess the PhoA family gene as in other *C. amphilecti*-like isolates (Table 1 and Appendix A, Figure 2).

It is evident that the PhoA coding sequences of the *Cobetia* strains are located in a common cluster with the LPS-detoxifying ALPs of humans and squid (Figure 3). The PhoA structural family of ALPs is predominantly associated with marine heterotrophic bacteria of *Bacteroidetes* and γ-*Proteobacteria*-neighbouring cyanobacteria populations [13,22], as well as microorganisms isolated from plant and animal microbiomes, such as symbiont (or pathogen) *Vibrio* spp. of planktonic crustaceans (Copepods), a symbiont (facultative pathogen) of human *E. coli*, a sugarcane root endophyte *Enterobacter roggenkampii* [29,63,64], and marine fish pathogens *Vibrio splendidus* [65] and *Moritella* sp. [66]. In addition, most eukaryotes have their own PhoA enzymes, with the exception of some plants, to modulate their microbiomes and tissue inflammation [9,10,11]. Pathogenic or symbiotic bacteria have probably evolved this strategy to evade host immunity, including the acquisition of the immunomodulatory PhoA alkaline phosphatase [66]. It follows that all PhoA alkaline phosphatases belong to a common homologous protein lineage [10] and are clustered into a separate branch of the phylogenetic tree (Figure 4).

In contrast to the species *C. amphilecti* (Table 1)*,* the ALP content of the subspecies *C. marina* and *C. pacifica* is restricted only by three genes of the families PhoD and PhoX (Table 1, Appendix A). Alkaline phosphatases of both PhoD and PhoX families are widely distributed in marine and soil bacteria. However, the PhoD family is more adapted to the lifestyle of soil bacteria, whereas the PhoX family is more characteristic of marine bacteria, including cyanobacteria, and for the soil microbial community in response to an increase in total organic matter content. In this regard, phoX genes dominate in the marine metagenomic databases [22,23]. The presence of PhoA family structures in the marine metagenomes may be an indicator of a large number of *Bacteroidetes* representatives, which are abundant during phytoplankton blooms [13,22]. In addition, most bacterial genomes are characterized by a single ALP family in the genome, whereas a single bacterium rarely has members of both the PhoX and PhoA, or PhoX/PhoA and PhoD families [22]. However, 35% of the owners of multiple paralogues of ALPs have at least one gene of the PhoA family, whereas 22 and 17% of the bacterial genomes have PhoX and PhoD genes, respectively [24]. The complete range of biological functions of paralogues encoding for alkaline phosphatase isoenzymes in an organism remains unknown. The differences in structural and, consequently, physicochemical properties of the ALP paralogues suggest their functional diversification during the assimilation of different organophosphates in P_i_-deficient conditions and other chemical parameters of the environment [13,24].

### 3.3. Biosynthetic Gene Cluster Analysis in the Genus Cobetia

The analysis of biosynthetic gene clusters (BGC) of the type strains of the genus *Cobetia* confirmed the species demarcation. Considering the main BGC inherent in the genus *Cobetia*, a high level of similarity of both the arrangement of genes in clusters and the sequences of individual genes was found. Despite the high BGC similarity between all *Cobetia* isolates, *C. litoralis* and *C. pacifica* correspond to the species *C. amphilecti* and *C. marina*, respectively [46,47], at the level of genus-specific BGC (Figure 5 and Figure 6). In particular, the ectoine BGC further differentiates the species *C. amphilecti* and *C. marina* by decreasing the sequence identity of the marker sensor genes encoding for diguanylate cyclase/phosphodiesterase, as well as a DUF3833 domain-containing protein of unknown function (Figure 5). The signalling pathway based on bacterial sensitivity to the concentration of cyclic dimeric GMP (c-di-GMP), a universal second messenger, controls cell growth and secondary metabolite production such as, apparently, the biosynthesis of ectoine in a high-salt medium [31,67].

The genus-specific BGC encoding for the Ni-siderophore synthesis genes was found to be slightly distinguishable between the closely related *Cobetia* species (Figure 6) by most genes. This indicates the high significance of iron uptake when bacteria grow under oxidative or iron deficiency conditions, particularly for Fe^2+/3+^ inclusion as a cofactor in the active sites of the specific enzymes, such as alkaline phosphatase/phosphodiesterase PhoD in *C. amphilecti* KMM 296 [31,45,48]. One hypothetical protein and sigma 70 family PNA polymerase factor might serve as the species-specific markers for demarcation of *C. ampilecti* and *C. pacifica* according to the new classification [47]. However, the genes of sigma 70 family PNA polymerase factor, aspartate aminotransferase, and monooxygenase related to this BGC, were not found in the *C. amphilecti* KMM 296 cluster, although they were extracted from the finished genome.

By contrast, the T1PKS BGC seems to be highly strain-specific (Figure 7).

However, several common key genes (capsular polysaccharide biosynthesis protein, SDR family NAD(P)-depended oxidoreductase, type I polyketide synthase, DsbA family oxidoreductase, etc.) encoding for the synthesis of extracellular polysaccharides of high conservativity are found in all genomes, and some divergence in capsular polysaccharide biosynthesis protein (61–65%) coincides with the new genome-based taxonomic classification for the genus *Cobetia* [47].

It has been determined that the strains *C. pacifica* KMM 3789^T^, KMM 3878, and *C. litoralis* KMM 3880^T^ produce sulfated O-polysaccharides (OPS) composed of trisaccharide repeating units and include D-glucose 3-sulfate and D-galactose 3-sulfate, D-galactose 2,3-disulfate, and 2-keto-3-deoxy-D-manno-octanoic acid 5-sulfate, respectively [68,69]. The *C. pacifica* capsular polysaccharides were found to exhibit an antiproliferative effect and suppress the colony formation of DLD-1 and MCF-7 cells in a different manner [68], whereas the LPS and O-deacetylated OPS from *C. litoralis* KMM 3880^T^ inhibited a colony formation of human melanoma SK-MEL-28 and colorectal carcinoma HTC-116 cells [69]. In this regard, it is of interest to study the polysaccharide structures of all *Cobetia* strains, particularly the *C. amphilecti* LPS, to determine the length and content of acyl chains present on lipid A. The ability of the PhoA enzyme CmAP to cleave other bacterial LPS and the *Cobetia* spp. LPS structures may shed light on its role in the *C. amphilecti* lifestyle, because the strains of *Moritella* spp. containing the lipid A with the highest amount of acyl chains (C16) and the PhoA family ALP were found to be immuno-silent for their host [66].

## 4. Conclusions

The highly active periplasmic CmAP-like alkaline phosphatases of *C. amphilecti* belong to the PhoA structural family, similar to the eukaryotic enzymes, for which LPS-dephosphorylating and detoxifying functions were confirmed. It is likely that some immuno-silent *Moritella*-like bacteria use the PhoA enzymes for imitation of the host ALP to manage other competing bacteria of the host’s microbiome, and to hide from the immune system. However, this hypothesis is still to be verified. Nevertheless, the abundance of divergent structural families of ALPs in the *Cobetia* isolates indicates their ability to survive in different ecological niches, with either high or low P_i_ content, because of their capability to degrade several types of organophosphates, probably including the energy metabolism cofactor NADPH. The P_i_-irrepressible CmAP-like *C. amphilecti* PhoA has been suggested as a participant in the maintenance of cellular redox balance, homeostasis, and reproductive cycle. This may provide other members of an ecological niche, including plants, with such essential macronutrients as phosphorus, as well as biocontrol. The ability to produce ectoine, a variety of sulfated exopolysaccharides, and the PhoA type alkaline phosphatases with LPS-dephosphorylating activity, ensure *Cobetia* species’ survival in highly salty and competitive conditions, as well as making them adaptable for use in agriculture, biotechnology and biomedicine.

## Figures and Tables

**Figure 1 microorganisms-12-00631-f001:**
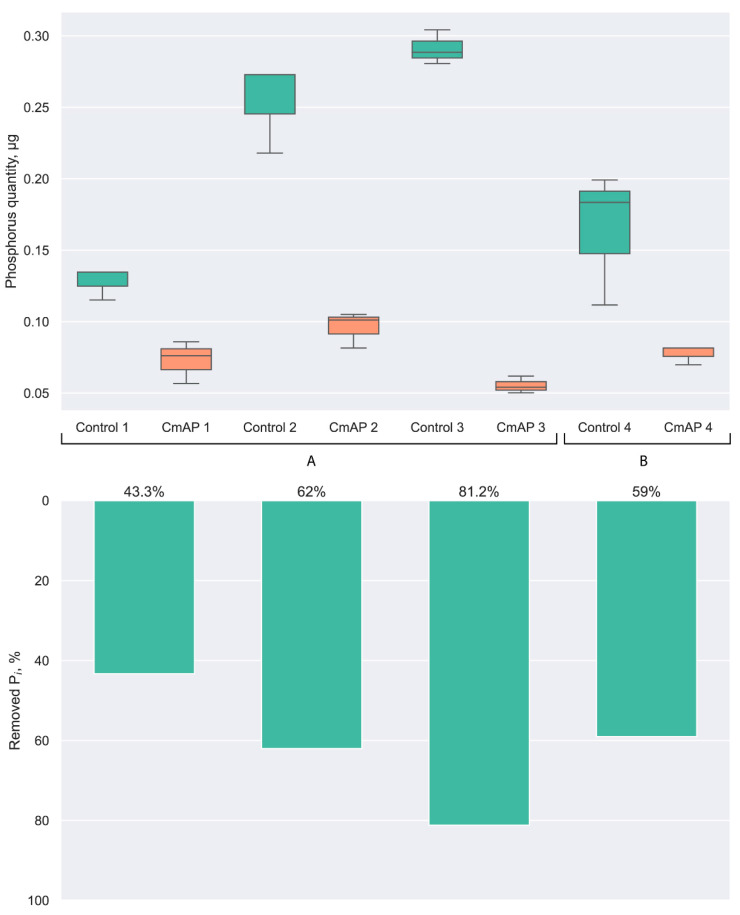
The phosphorus quantity (µg) in *E. coli* 055:B5 LPS samples 1–4 (70 µL, 0.2 mg/mL) before (control, green) and after (CmAP, orange) treatment with the recombinant alkaline phosphatase CmAP (0.012 mg/mL, 2300 *p*-NPP U/mg) for 1 h at 37 °C, with the use of LPS samples dissolved in the buffer A (0.1 M Tris-HCl buffer, 0.1 M KCl, pH 7.7): 1—at 24 °C for 12 h; 2—at 37 °C for 12 h; 3—at 24 °C for 12 h with the addition of TEA (1µL/mL) in buffer B (0.1 M Tris-HCl buffer, 0.1 M KCl, pH 10.0); 4—at 24 °C for 12 h. The average quantity of hydrolyzed inorganic phosphate in each sample is shown below as % of the removed phosphorus (P_i_) after dialysis compared to the control.

**Figure 2 microorganisms-12-00631-f002:**
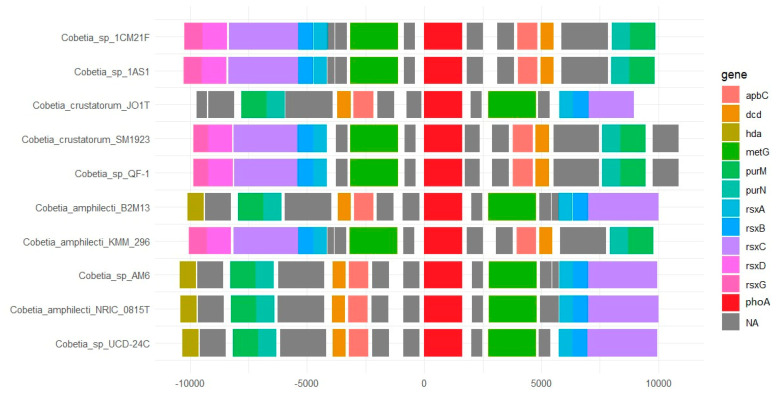
The putative biosynthesis gene cluster for the PhoA family alkaline phosphatase in the strains of *C. amphilecti* (1CM21F, 1AS1, QF-1, B2M13, KMM 296, AM6, NRIC 0815^T^, and UCD-24C) and *C. crustatorum* (JO1^T^, SM1923). The distances between coding sequences upstream and downstream of the start codon for PhoA (from 0 to 10,000) in a chromosome are in codons (amino acids). For functional annotation verification, the alignment was performed for all protein sequences included in the gene cluster. NA (not applicable) means that the functional annotation for that gene is absent in the genome annotation (GTF).

**Figure 3 microorganisms-12-00631-f003:**
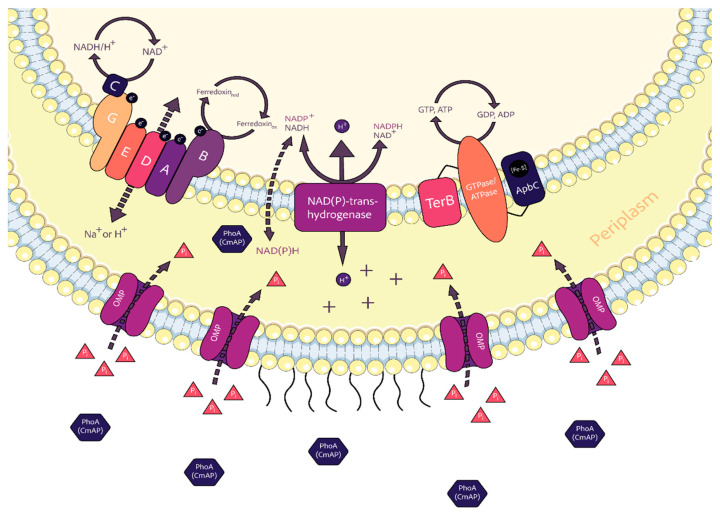
The putative PhoA alkaline phosphatase involvement in the maintenance of cellular redox balance and homeostasis in the marine bacteria *Cobetia* spp.: PhoA(CmAP) (dark blue hexagon)—alkaline phosphatase of the family PhoA, matured in the periplasm, to be released into the extracellular environment and non-specific cleavage of dissolved phosphates; P_i_ (red triangles)—inorganic phosphate anions, produced by PhoA to be uptaken by mass transfer across the outer membrane through porins (OMP, violet channels), buffered by chemiosmotic cations of the membrane potential (+) in the periplasm and imported from the buffered stock through the cytoplasmic membrane; ABCDEG (colourful shapes)—electron transport complex subunits Rsx coupled with NAD(H)/NAD^+^ and Ferredoxin_reduced_/_oxidized_ interconversion; NAD(P)(^+^)-transhydrogenase (purple rectangle)—DUF3465 domain-containing protein related to membrane-bond Si-specific NAD(P)(^+^)-transhydrogenase pumping protons (H^+^) and generating inter-reversible redox couples NAD(H) and NADP(H) by respiration and ATP hydrolysis; NAD(P)H (in periplasm)—extracellular nicotinamide adenine dinucleotide (phosphate) reduced which might be used by PhoA as a substrate for dephosphorylation and production of NAD(H); GTPase/ATPase (orange oval)—protein complex catalyzing the decomposition of ATP and GTP into ADP or GDP for energy-dependent cell processes; TerB (red rectangle)—GTP-dependent protein family related to antiviral defence and repairing xenobiotic-induced DNA damage; ApbC (duck blue rectangle)—ATP-dependent [Fe-S] carrier protein. LPS are depicted as filaments on the outer membrane of the cell.

**Figure 4 microorganisms-12-00631-f004:**
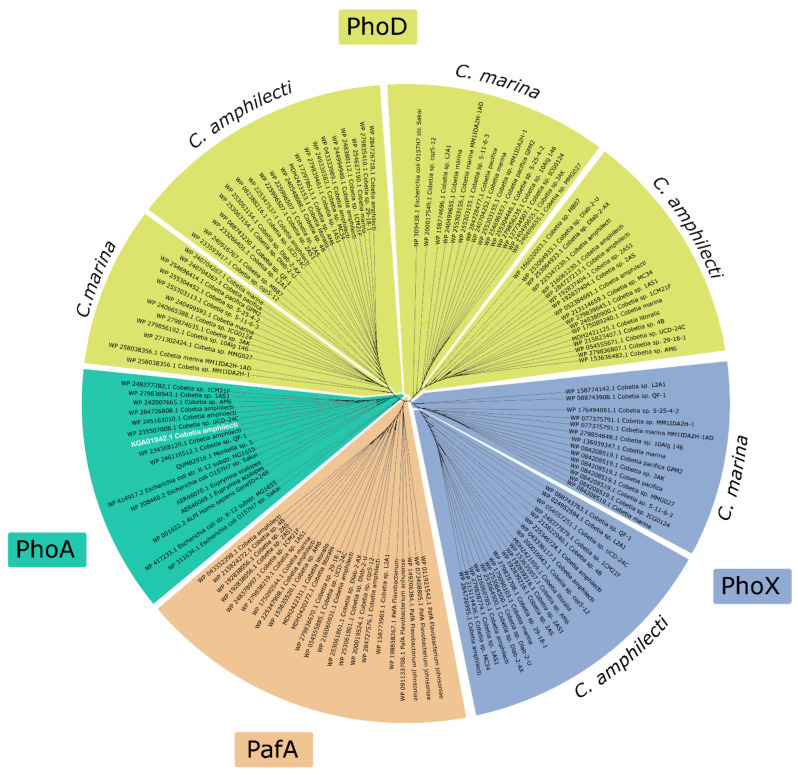
Phylogenetic tree based on the similarity of full-length amino acid sequences of the *Cobetia* spp. alkaline phosphatases constructed using MEGA 11 software by the maximum likelihood method. GenBank database access numbers are indicated before the species titles of the strains. Bootstrap values at the nodes of the bootstrap consensus tree constructed from 1500 samples are at least 50%. Alkaline phosphatase structures are grouped into protein families: alkaline phosphatase PhoD; alkaline phosphatase PhoA; alkaline phosphatase PafA; and alkaline phosphatase PhoX. *C. amphilecti* KMM 296 alkaline phosphatase CmAP (accession no. KGA01942) within the PhoA family cluster is marked in white.

**Figure 5 microorganisms-12-00631-f005:**
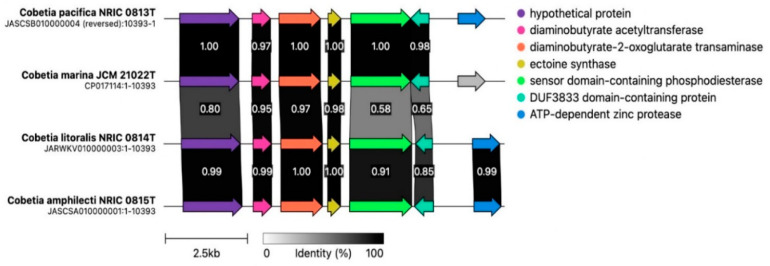
The synteny of ectoine BGC in the type strains of the *Cobetia* species. The different identity between the genes is reflected by a gradient of grey and the values of identity; identical genes are in black. The grey arrow indicates an identity below threshold of 0.3 (30%).

**Figure 6 microorganisms-12-00631-f006:**
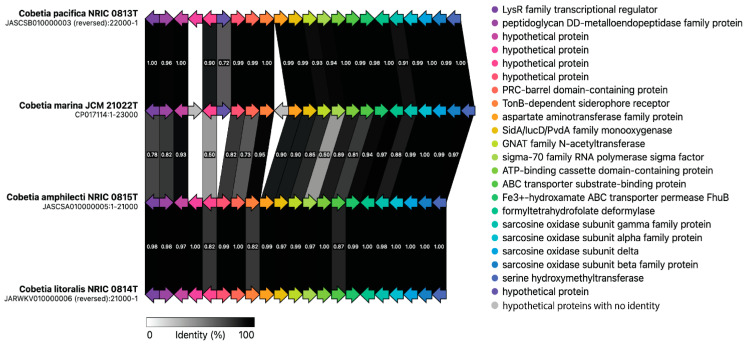
The synteny of Ni-siderophore BGC in the type strains of the *Cobetia* species. The functional annotation for each gene is presented in the panel on the right. The differences between the genes are reflected by a gradient of grey and the values of identity (%);identical genes are in black. The grey arrows indicate an identity below threshold of 0.3 (30%).

**Figure 7 microorganisms-12-00631-f007:**
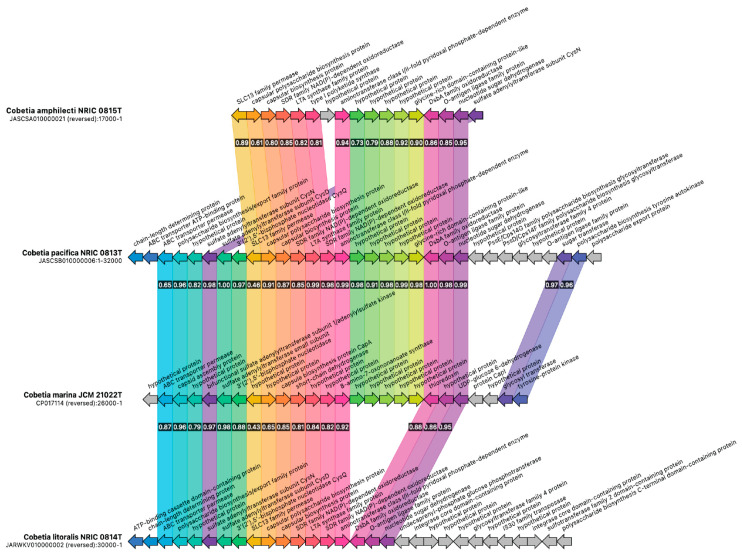
The synteny of T1PKS BGC encoding for extracellular polysaccharide production in the type strains of the genus *Cobetia.* The singletons are marked by grey arrows.

**Table 1 microorganisms-12-00631-t001:** The content and distribution of alkaline phosphatase families in the *Cobetia* spp. type strains and isolates.

Original Strain Title	Actual Species *	Accession ID	ID Protein	ALP Family	Isolation Source
*C. marina* JCM 21022^T^	*C. marina*	GCF_001720485.1	WP_240499655.1	PhoD	Littoral water sample (USA: Woods Hole, MA, USA)
			WP_084208519.1	PhoX	
			WP_240499593.1	PhoD	
*C. pacifica* NRIC 0813^T^	*C. marina*	GCA_030010515.1	WP_284728477.1	PhoD	Sandy sediment (Russia: The Sea of Japan)
			WP_240704267.1	PhoX	
			WP_084208519.1	PhoX	
*C. litoralis* NRIC 0814^T^	*C. amphilecti*	GCF_029846315.1	WP_249330383.1	PhoD	Sandy sediment (Russia: The Sea of Japan)
			WP_279830791.1	PafA	
			WP_279833222.1	PhoX	
			WP_279832006.1	PhoD	
*C. amphilecti* NRIC 0815^T^ **	*C. amphilecti*	GCA_030010415.1	WP_284726718.1	PhoD	The finger sponge *A. digitatus* (The Sea of Okhotsk, Sakhalin Island, Piltun Bay)
			WP_284726808.1	PhoA **	
			WP_284726995.1	PhoX	
			WP_284727213.1	PhoD	
			WP_284727576.1	PafA	
*C. amphilecti* KMM 296 **	*C. amphilecti*	GCF_000754225.1	WP_043332298.1	PafA	Coelomic fluid of mussel *C. grayanus* (Russia: The Sea of Japan)
			WP_043333989.1	PhoD	
			WP_245163010.1	PhoA **	
			WP_043336117.1	PhoX	
			WP_052384691.1	PhoD	
*C. amphilecti* B2M13 **	*C. amphilecti*	GCF_018860945.1	WP_244994940.1	PhoD	Alginate 40–100 m particle (artificial) (USA: Canoe Beach)
			WP_234368129.1	PhoA **	
			WP_216060785.1	PhoX	
			WP_216060931.1	PafA	
			WP_216061230.1	PhoD	
*Cobetia* sp. 1AS1 **	*C. amphilecti*	GCF_029846435.1	WP_279838219.1	PafA	Coastal seawater (Russia: the Sea of Japan, Vostok Bay)
			WP_279838543.1	PhoA **	
			WP_279838774.1	PhoX	
			WP_279839461.1	PhoD	
			WP_279839645.1	PhoD	
*Cobetia* sp. 1CM21F **	*C. amphilecti*	GCF_023161745.1	WP_248377282.1	PhoA **	Sea cave (Portugal: Algarve)
			WP_248377879.1	PhoX	
			WP_248378997.1	PafA	
			WP_248380112.1	PhoD	
			WP_248380900.1	PhoD	
*Cobetia* sp. AM6 **	*C. amphilecti*	GCF_009617955.1	WP_172978613.1	PhoD	Exterior surface of the shell of an abalone sold in a fish market (Tokyo, Japan)
			WP_153635520.1	PafA	
			WP_153635957.1	PhoX	
			WP_242007665.1	PhoA **	
			WP_153636482.1	PhoD	
*Cobetia* sp. UCD-24C **	*C. amphilecti*	GCF_001306765.1	WP_082388216.1	PhoD	Seagrass *Zostera* sp. sediment
			WP_054555671.1	PhoD	
			WP_054555885.1	PafA	
			WP_235507008.1	PhoA **	
			WP_054557251.1	PhoX	
*C. crustatorum* JO1^T^ **	*C. crustatorum*	GCF_000591415.1	WP_248623642.1	PhoA	Fermented shrimp (South Korea: Daejeon)
			WP_282705494.1	PhoD	
			WP_282705495.1	PhoD	
			WP_024952594.1	PhoX	
*C. crustatorum* SM1923 **	*C.* *crustatorum*	GCF_007786215.1	WP_144726746.1	PhoD	Surface seawater (Kongsfjorden, Arctic)
			WP_088743763.1	PhoD	
			WP_144727163.1	PhoX	
			WP_246116512.1	PhoA **	
			WP_144728015.1	PhoD	

* According to the new classification of Nedashovskaya et al., 2024 [47]. ** The strains containing the PhoA family.

**Table 2 microorganisms-12-00631-t002:** Comparative analysis of the PhoA family structures found in the *Cobetia* genomes.

Strain_ID Genome	Mismatch
*Cobetia_amphilecti*_B2M13_GCF_018860945.1	240 F > Y, 308 F > Y
*Cobetia_amphilecti*_NRIC_0815T_GCA_030010415.1	240 F > Y, 308 F > Y, 372 E > G, 399 A > T
*Cobetia_crustatorum*_JO1T_GCF_000591415.1	Down start (+45 amino acids) missing key position 21D, Tblastn_Identities 413/466 (89%)
*Cobetia_crustatorum*_SM1923_GCF_007786215.1	Tblastn_Identities 456/511 (89%)
*Cobetia*_sp_1AS1_GCF_029846435.1	240 F > Y, 308 F > Y
*Cobetia*_sp_1CM21F_GCF_023161745.1	240 F > Y, 308 F > Y
*Cobetia*_sp_AM6_GCF_009617955.1	240 F > Y, 308 F > Y, 419 A > V, 422 P > L
*Cobetia*_sp_UCD-24C_GCF_001306765.1	240 F > Y, 308 F > Y, 390 G > D

## Data Availability

Data available in a publicly accessible repository.

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
