# Peer review of "LPS-Dephosphorylating Cobetia amphilecti Alkaline Phosphatase of PhoA Family Divergent from the Multiple Homologues of Cobetia spp."

_microorganisms, 2024, doi:10.3390/microorganisms12030631_

Round 1

Reviewer 1 Report (Previous Reviewer 2)

Comments and Suggestions for Authors

The resubmitted manuscript described the study of a Cobetia amphilecti KMM 296 alkaline phosphatase (CmAP) belonging to the PhoA family capable of dephosphorylation of Escherichia coli lipopolysaccharides (LPS). The alkaline phosphatase sequences were analyzed in 36 Cobetia genomes, and the results showed that CmAP and its homologs from nine strains clustered together with the human and squid LPS-detoxifying enzymes. It was assumed that the whole-genome-based approach was helpful in the taxonomic classification of the Cobetia genus. The discussion on smooth-LPS (S-LPS) and rough-LPS (Ra-LPS) chemotypes was removed in the current manuscript, making it more significant than the previous one. Though most questions were solved, due to insufficient and limited data available for review, the manuscript needs to be further evaluated for the journal Microorganisms. Below are the main reasons.

1.     Section 2.1.

It was suggested that the SDS-PAGE images should be provided because the procedures for expressing and purifying recombinant CmAP were not the same as those presented in the previous study (10.1007/s10126-014-9601-0). It can be provided as the supplementary material. However, the data was not provided.

2.     Section 3.3. Figure 4.

The arrow in grey was not indicated.

3.     The previous questions included that the genus-specific BGC encoding for the Ni-siderophore synthesis genes was found to be indistinguishable from the Cobetia isolates (Figure S2). In fact, it was Figure S3 that also needs to be further discussed. How do we define indistinguishable because it showed variations in Figure S3? In addition, why were only 25 genomes analyzed? However, it cannot be further evaluated because no supplementary data was available.

4.     Figure S1.

It was previously suggested that a full-screen image capture was not a good way to show the result of multiple sequence alignment. However, it cannot be further evaluated because no supplementary data was available.

5.     Figure 2.

The diagram included several other Cobetia sp. The legend should be corrected. In addition, the definition of the scale below the diagram needs to be modified. What is the meaning of “0”? Is “+1” better to show the start of the phoA gene?

6.     Figure 5.

Were the arrows in grey all nucleotide sugar dehydrogenases? Didn’t they show high conservatives among all genomes?

7.     Tables S1 and S2 were not found.

Comments on the Quality of English Language

Minor editing of English language required.

Author Response

The resubmitted manuscript described the study of a Cobetia amphilecti KMM 296 alkaline phosphatase (CmAP) belonging to the PhoA family capable of dephosphorylation of Escherichia coli lipopolysaccharides (LPS). The alkaline phosphatase sequences were analyzed in 36 Cobetia genomes, and the results showed that CmAP and its homologs from nine strains clustered together with the human and squid LPS-detoxifying enzymes. It was assumed that the whole-genome-based approach was helpful in the taxonomic classification of the Cobetia genus. The discussion on smooth-LPS (S-LPS) and rough-LPS (Ra-LPS) chemotypes was removed in the current manuscript, making it more significant than the previous one. Though most questions were solved, due to insufficient and limited data available for review, the manuscript needs to be further evaluated for the journal Microorganisms. Below are the main reasons.

  1. Section 2.1.

It was suggested that the SDS-PAGE images should be provided because the procedures for expressing and purifying recombinant CmAP were not the same as those presented in the previous study (10.1007/s10126-014-9601-0). It can be provided as the supplementary material. However, the data was not provided.

- The CmAP expression and purification results were presented in the supplemental Figure S1, which zipped with the main manuscript. Probably, it will be available.

  1. Section 3.3. Figure 4.

The arrow in grey was not indicated.

- The gray arrow indicates an identity below threshold of 0.3. (Figure 5, Line 625). Thanks.

  1. The previous questions included that the genus-specific BGC encoding for the Ni-siderophore synthesis genes was found to be indistinguishable from the Cobetiaisolates (Figure S2). In fact, it was Figure S3 that also needs to be further discussed. How do we define indistinguishable because it showed variations in Figure S3? In addition, why were only 25 genomes analyzed? However, it cannot be further evaluated because no supplementary data was available.

- The Figure was restricted to only four type strains, similarly to other Figures of clusters. Indeed, there are small differences in two genes that support the division of the genus Cobetia into two species (Figure 6, Lines 626-629, Lines 635-640). Thank you for insisting. The Figure S3 was removed.

  1. Figure S1.

It was previously suggested that a full-screen image capture was not a good way to show the result of multiple sequence alignment. However, it cannot be further evaluated because no supplementary data was available.

- The Figure S1 containing the multiple sequence alignment is renamed now as Figure S2 due to the addition of the gel-electrophoresis image at the Figure S1 according to the reviewer's recommendation. Now we have attached the file alone. Probably, it will be available.   

  1. Figure 2.

The diagram included several other Cobetia sp. The legend should be corrected. In addition, the definition of the scale below the diagram needs to be modified. What is the meaning of “0”? Is “+1” better to show the start of the phoA gene?

- The diagram indeed includes only the strains of Cobetia amphilecti and Cobetia crustatorum. The alkaline phosphatase PhoA is absent in Cobetia litoralis, Cobetia pacifica and Cobetia marina as presented in the Tables 1 (column 2) and Table S2. The legend has been clarified accordingly. Thanks. (Figure 2, Lines 454-460).

  1. Figure 5.

Were the arrows in grey all nucleotide sugar dehydrogenases? Didn’t they show high conservatives among all genomes?

- Each gray arrow is designated now in the Figure 7. However, they are not identical. The conservative genes are highlighted by synteny in all Figures.  (Lines 642-643)

  1. Tables S1 and S2 were not found.
  • Probably, they are available now.

Reviewer 2 Report (New Reviewer)

Comments and Suggestions for Authors

Please provide mechanism of alkaline phosphatase and the genes involves by graph/figure

Comments on the Quality of English Language

Need to improve

Author Response

This manuscript report about alkaline phosphatase (ALP) from marine bacterium Cobetia amphilecti KMM296.The author did functional characterization of the enzymes. The author also analyzed diversity of the enzymes in 36 available Cobetia genomes as well as other species. The study is very interesting but more information is needed before publication. I would suggest the author to provide one figure that describe the mechanism of alkaline phosphatase and the genes involved in the mechanism. Minor comment, Abstract need to rewrite. Include objective, and application of the finding

  • The Figure 3 describes the putative mechanism of the alkaline phosphatase PhoA and the genes involved in the mechanism as suggested by the Reviewer. We have to include the additional references. Thanks. (Lines 479-541).
  • The objective and application of the finding have been included as suggested by the Reviewer. Thanks. (Lines 19-22, 37-38).
  • The English gramma was kindly checked by the native speaker suggested by the Reviewer. 

Round 2

Reviewer 1 Report (Previous Reviewer 2)

Comments and Suggestions for Authors

The revised resubmitted manuscript described the study of a Cobetia amphilecti KMM 296 alkaline phosphatase (CmAP) belonging to the PhoA family capable of dephosphorylation of Escherichia coli lipopolysaccharides (LPS). The alkaline phosphatase (ALP) sequences were analyzed in 36 Cobetia genomes, and the results showed that CmAP and its homologs from nine strains clustered together with the human and squid LPS-detoxifying enzymes. It was assumed that the Cobetia sp. and the ALPs were useful in agriculture, biotechnology, and biomedicine. It was noted that most questions were solved according to the comments and suggestions. It’s now recommended for the journal Microorganisms after the following questions are addressed.

1.     General rules of gene and protein formats should be followed. The manuscript contained mixed gene and protein names, which should be examined and corrected carefully.

1.1 Gene symbols are italicized, all lowercase. For example, phoA gene.

1.2 Protein designations are the same as the gene symbol, but the first letter is only in uppercase and not italicized. For example, PhoA protein.

2.     Section 2.1 and Figure S1.

The SDS-PAGE images were provided but still need improvements.

2.1  Figure S1A. Lanes 3-5 and 7-10 represented the cell extracts from the recombinant colonies E. coli Rosetta (DE3)/Pho40 after IPTG-induced expression. However, the molecular weights of the recombinant proteins were not the same, especially in lanes 3 and 10. It needs an explanation in the text.

2.2  It was stated that a metal affinity resin was used to purify the recombinant protein and then cleaved by an enteropeptidase and applied to an ion exchange sorbent. It is suggested that the SDS-PAGE images be added: 1) elution from a metal affinity resin to show the affinity-purified uncleaved recombinant protein; 2) recombinant protein after cleavage by the enteropeptidase.

3.     Figure 2.

1.1   The legend described one of C. amphilecti strains, RMM 296. Wasn’t it KMM_296?

1.2   What was the “OX” axis?

1.3   The genome annotation file format is GTF (uppercase).

4.     Section 3.3. Figures 5 and 6.

It was described that the gray arrow indicates an identity below the threshold of 0.3. However, the identity (%) was used in the figures. It is suggested that 30% should be used instead of 0.3.

Author Response

I am grateful to the Reviewers for their careful work on the manuscript, which has improved its quality. 

All recent changes in response to the reviewer's requirements are highlighted in colour and indicated in the response below.

Comments and Suggestions for Authors:

The revised resubmitted manuscript described the study of a Cobetia amphilecti KMM 296 alkaline phosphatase (CmAP) belonging to the PhoA family capable of dephosphorylation of Escherichia coli lipopolysaccharides (LPS). The alkaline phosphatase (ALP) sequences were analyzed in 36 Cobetia genomes, and the results showed that CmAP and its homologs from nine strains clustered together with the human and squid LPS-detoxifying enzymes. It was assumed that the Cobetia sp. and the ALPs were useful in agriculture, biotechnology, and biomedicine. It was noted that most questions were solved according to the comments and suggestions. It’s now recommended for the journal Microorganisms after the following questions are addressed.

  1. General rules of gene and protein formats should be followed. The manuscript contained mixed gene and protein names, which should be examined and corrected carefully.

1.1 Gene symbols are italicized, all lowercase. For example, phoA gene.

Reply: - The gene names have been checked out through the manuscript, for the exception of the places where the genes are designated by a program.

1.2 Protein designations are the same as the gene symbol, but the first letter is only in uppercase and not italicized. For example, PhoA protein.

Reply: - The protein designations have been checked out. Thanks.

  1. Section 2.1 and Figure S1.

The SDS-PAGE images were provided but still need improvements.

2.1  Figure S1A. Lanes 3-5 and 7-10 represented the cell extracts from the recombinant colonies E. coli Rosetta (DE3)/Pho40 after IPTG-induced expression. However, the molecular weights of the recombinant proteins were not the same, especially in lanes 3 and 10. It needs an explanation in the text.

Reply: - Figure S1 A contained two variants of the recombinant protein CmAP, including one of the CmAP artificial hybrid, with 17 kDa higher weigh than CmAP alone. The lanes 4 and 9 were related to the recombinant CmAP alone. Therefore, we have removed the lanes 3, 5-10 unrelated to this manuscript (Figure S1 A). Thanks for your attention.

2.2  It was stated that a metal affinity resin was used to purify the recombinant protein and then cleaved by an enteropeptidase and applied to an ion exchange sorbent. It is suggested that the SDS-PAGE images be added: 1) elution from a metal affinity resin to show the affinity-purified uncleaved recombinant protein; 2) recombinant protein after cleavage by the enteropeptidase.

Reply: -It has been included (Figure S1 B). Thanks.

  1. Figure 2.

1.1   The legend described one of C. amphilecti strains, RMM 296. Wasn’t it KMM_296?

Reply: - Of course. It has been corrected. Thanks.

1.2   What was the “OX” axis?

Reply: - It has been removed and corrected (Figure 2. Lines 455-456). Thanks.

1.3   The genome annotation file format is GTF (uppercase).

Reply: - It has been altered as suggested by the Reviewer (in green). Thanks.

  1. Section 3.3. Figures 5 and 6.

It was described that the gray arrow indicates an identity below the threshold of 0.3. However, the identity (%) was used in the figures. It is suggested that 30% should be used instead of 0.3.

  • Reply: - The values were assigned by the program, therefore 30% are indicated in the figure caption next to the value 0.3 in brackets (Lines 623, 627).

This manuscript is a resubmission of an earlier submission. The following is a list of the peer review reports and author responses from that submission.

Round 1

Reviewer 1 Report

Comments and Suggestions for Authors

The article is written at a level that is sufficient to understand it, all parts are present and appropriate. Although neither the title nor the abstract reflects the fact that the article is a mere compilation of two independent texts, which exist separately, not wanting to unite despite the efforts of the authors...
The abstract reviews the text of the article and its main points.
The introduction is extensive and provides an overview sufficient to set out the problem and covers the research methods used.
line 45: to me "alcohol" as a reaction product looks, although understandable, somewhat provocative!
The methods are described in detail but a bit snaggy!
line 122: Novagen
line 133: here "4000×g rpm" and further down convert to acceleration not RPM for centrifugations
line 166: the equation looks childish, please format it according to common standards. It's also a good idea to link this paragraph, since it's a standard assay

Results and discussion
The protein used in the work is rather well purified (it should be at least) judging by the methods. Nevertheless, the absence of a picture of the stained gel after SDS-PAGE according to Laemmli is questionable: where is it? It is not bad to put there the control of cells without recombinant plasmid. Also would like to see the same control in all reactions to be sure that the E. coli fraction does not have the mentioned activity/protein.

Line 300: figures. the use of diagrams is controversial! Wouldn't it be better to use tables? The captions are hardly readable!
Line 470: the figure should be reconsidered! At the present moment everything on it is not readable!

Reviewer 2 Report

Comments and Suggestions for Authors

The manuscript described the study of a Cobetia amphilecti KMM 296 alkaline phosphatase (CmAP) belonging to the PhoA family capable of dephosphorylation of Escherichia coli lipopolysaccharides (LPS), especially the smooth-LPS (S-LPS) chemotype compared to rough-LPS (Ra-LPS) chemotype. The alkaline phosphatase sequences were analyzed in 36 Cobetia genomes, and the results showed that CmAP and its homologs from nine strains clustered together with the human and squid LPS-detoxifying enzymes. It was assumed that the whole-genome-based approach was useful for the taxonomic classification of Cobetia genus. Due to insufficient and limited data available for review, the manuscript is not suitable for the journal Microorganisms. Below are the main reasons.

1.     Section 2.1. The SDS-PAGE images should be provided because the procedures for expression and purification of recombinant CmAP were not the same as those presented in the previous study (10.1007/s10126-014-9601-0). It can be provided as the supplementary material.

2.     Section 2.3. Two different LPS chemotypes were used as substrates for the enzymatic reaction. One was the smooth-LPS (S-LPS) chemotype, and the other was the rough-LPS (Ra-LPS) chemotype. The full names of chemotypes should be described first, and the abbreviations should be described later. In addition, the source of S-LPS was incorrect.

3.     Section 2.3. It lacked unit (hours) for sample No. 5.

4.     Section 3.1. It was described that the amount of detectable phosphorus in the S-LPS sample solubilized in 0.1 M Tris-HCl buffer, pH 7.7, increased 6-fold after reducing the LPS concentration from 1 to 0.2 mg/mL and increasing the incubation temperature to 37 °C (sample No. 2) (Figure 1a). However, the data of 1 mg/mL was not available here. A 6-fold increase was also overexplained.

5.     As above, it was described that it was significantly increased after the addition of triethylamine (TEA) and, consequently, alkalization incubation medium to pH 10.0 (sample No. 3) compared to the LPS sample No. 1 (S-LPS incubation at 24°C and without an addition TEA). However, it did not describe the concentration of TEA used or explain why TEA was used to alkalize the solution. An additional control is required to show the effects of TEA on the dissolution.

6.     Section 3.1 and Figure 1b. A different DEA (Diethanolamine?) was used to show the dependence of the determined amount of phosphorus in the Ra-LPS samples (â„– 2, 3, and 5) on different dissolution conditions. It did not describe why DEA experiments were conducted on Ra-LPS. Were the results of Figure 1a and 1b comparable?

7.     As above, K3 – (Ra-LPS sample â„– 3) in Figure 1b used Tris-HCl again and alkalized by TEA. It also needs an explanation.

8.     Section 3.1 and Figure 1. The reaction duration was different for S-LPS and Ra-LPS. Were the results of Figure 1a and 1b comparable?

9.     Figure 2 and Figure 1a were reductants, and the standard deviations were different.

10.  If Figure 1b was assumed to be comparable, the results after treatment with recombinant alkaline phosphatase CmAP have to be shown for further comparison.

11.  Section 3.1 It was described that the reaction time at the optimal conditions should be at least 30 min at the protein concentration 0.0003 - 0.024 mg/mL of the CmAP preparation with the specific pNPP-dephosphorylation activity 2300 U/mg protein. The experimental data have to be presented.

12.  As above, it was described that the enzyme CmAP is able to dephosphorylate LPS almost completely if it has the complete structure (S-form) and is pre-incubated in a buffer with a pH value ≥10.0 followed by incubation for 12 h at 37°C (Figure 2). However, it was overexplained because no other alkaline condition was performed. In fact, Figure 2 did not show the result of pre-incubation in a buffer with a pH value ≥10.0 followed by incubation for 12 h at 37°C. Instead, 24°C was used for this condition.

13.  Tables 1 and 2. Six additional Cobetia genomes that had PhoA family in Table 2 were suggested to be included in Table 1.

14.  Figure 3. The phylogenetic distance has to be shown in the figure.

15.  Section 3.3. It was described that the ectoine BGCs allow the division of the species C. amphilecti and C. marina by decreasing the sequence identity of the marker sensor genes encoding for diguanylate cyclase and phosphodiesterase (Figure 4). However, it was not easy to judge the sequence similarity according to the illustration because they were reflected only by the gray gradient. In addition, DUF3833 domain-containing protein and ATP-dependent zinc protease may show a similar situation. Why were they not included and discussed?

16.  It was described that the genus-specific BGC encoding for the Ni-siderophore synthesis genes was found to be indistinguishable from the Cobetia isolates (Figure S2). In fact, it was Figure S3 that also needs to be further discussed. How do we define indistinguishable because it showed variations in Figure S3? In addition, why were only 25 genomes analyzed?

17.  The use of C. amphilecti CmAP for the development of anti-inflammatory drugs needs to be further discussed at the end of Section 4 or in Section 5. Conclusions to avoid overinterpretation in the abstract and at the end of Section 1. Introduction.

18.  Figure S1. A full-screen image capture was not a good way to show the result of multiple sequence alignment.

19.  Figure S2. It needs further explanation to show the meaning of tree scale and 0-5, the genomes included for building the phylogenetic tree, and the iTOL tool used for the analysis.

Comments on the Quality of English Language

Moderate editing of the English language required